# A Set of Plasmid-Based Modules for Easy Switching of C-Terminal Epitope Tags in *Saccharomyces cerevisiae*

**DOI:** 10.3390/microorganisms9122505

**Published:** 2021-12-03

**Authors:** Hiroki Hayashi, Tsutomu Kishi

**Affiliations:** College of Engineering, Nihon University, Koriyama, Fukushima 963-8642, Japan; cehr18001@g.nihon-u.ac.jp

**Keywords:** epitope switch, epitope tagging, one-step PCR, MX6 module, TAP tag, *Saccharomyces cerevisiae*

## Abstract

Epitope tagging is a powerful strategy for analyzing the functions of targeted proteins. The use of this strategy has become more convenient with the development of the epitope switch, which is another type of epitope tagging designed to convert the previously tagged epitopes on the chromosome to other epitopes of interest. Various modules for C-terminal epitope switching have been developed and amplified using the one-step polymerase chain reaction (PCR) method before transformation. However, PCR amplification occasionally generates mutations that affect the fidelity of epitope switching. Here, we constructed several plasmids to isolate modules for epitope switching through digestion by restriction enzymes. The isolated modules contained DNA sequences for homologous recombination, various epitopes (13×Myc, 6×HA, GFP, Venus, YFP, mCherry, and CFP), and a transformation marker (*Candida glabrata LEU2*). The restriction enzyme-digested plasmids were used to directly transform the cells for epitope switching. We demonstrate the efficient and accurate switching of the MX6 module-based C-terminal tandem affinity purification tags to each aforementioned epitope. We believe that our plasmids can serve as powerful tools for the functional analysis of yeast proteins.

## 1. Introduction

Epitope tagging is a powerful strategy for analyzing the functions of target proteins. Because of the existence of an efficient homologous recombination mechanism in *Saccharomyces cerevisiae*, the epitope of interest can be integrated into a desired position on the chromosome [1]. Transformation constructs for epitope tagging can be prepared using the one-step polymerase chain reaction (PCR) method [2,3,4,5]. In this method, insertion modules that are composed of DNA sequences encoding an epitope and transformation markers are used as PCR templates. Long oligonucleotides are used as primers, which have 40-nucleotide sequence identities with the target recombination sites as well as 20-nucleotide sequence identities with the template plasmids. Various modules with different epitopes were constructed and used as templates for the one-step PCR method. Currently, tandem affinity purification (TAP)-tagged yeasts in which almost all genes are C-terminally TAP-tagged were constructed [6] and are commercially available.

Epitope switching is a type of epitope tagging that is designed to convert a previously tagged epitope on the chromosome to another epitope of interest. Sung et al. [7] developed plasmids with several insertion modules so that the one-step PCR method could be applied for epitope switching. This PCR-based method of epitope switching has several advantages over epitope tagging using one-step PCR. The epitope switch can be more efficient than epitope tagging because the insertion modules have relatively long homology regions. In addition to being efficient, it is more economical than the conventional method. The epitope switching method developed by Sung et al. [7] uses short common primer pairs, although epitope tagging using one-step PCR requires long gene-specific primer pairs. Sung et al. [7] efficiently switched the MX6 module-based C-terminal TAP tags with various epitopes. Moreover, they suggested that the modules for transformation could be generated using the restriction digestion of their plasmids. Direct use of the restriction enzyme-digested plasmids could make the epitope switching more convenient because it does not require the use of PCR to generate transformation modules [7]. Alexandrov et al. [8] developed the pAM781D plasmid, which is capable of generating the modules necessary for switching the C-terminal GFP (S65T) to an RFP tag using restriction digestion, and they demonstrated this conversion from *SSA2-GFP-His3MX* to *SSA2-RFP*-*Ogataea polymorpha*
*URA3*.

In the present study, we constructed a set of plasmids bearing modules for epitope switching using restriction digestion. Using these plasmids, we demonstrated the efficient and accurate switching of the MX6 module-based C-terminal TAP tags to one of the following epitopes: 13×Myc, 6×HA, GFP, Venus, YFP, mCherry, and CFP. We constructed our plasmids on the basis of the study conducted by Sung et al. [7], but we used *Candida Glabrata LEU2* (*cgLEU2*) as a transformation marker instead of *Kluyveromyces lactis URA3* (*KIURA3*), which expands the utility of the epitope switch. Therefore, we think that our plasmids can serve as useful tools for analyzing protein function.

## 2. Materials and Methods

### 2.1. Yeast Strains and Media

The *Saccharomyces cerevisiae* strains used in the present study are listed in Table 1 and are derived from the BY4741 strain, except for BY25730, which is a derivative of W303. The cells were grown in YPD medium (1% yeast extract, 2% peptone, and 2% glucose), YPR medium (1% yeast extract, 2% peptone, and 2% raffinose), YPRG medium (1% yeast extract, 2% peptone, 2% raffinose, and 2% galactose), SD medium (1.7% yeast nitrogen base without amino acids and ammonium sulfate, 0.5% ammonium sulfate, and 2% glucose) with appropriate amino acids, SRaff medium (1.7% yeast nitrogen base without amino acids and ammonium sulfate, 0.5% ammonium sulfate, and 2% raffinose) with appropriate amino acids, and SRaffGal medium (1.7% yeast nitrogen base without amino acids and ammonium sulfate, 0.5% ammonium sulfate, 2% raffinose, and 2% galactose) with appropriate amino acids. To solid media, we added 2% agarose. Yeast transformation was performed as previously described by Gietz and Schiestl [9].

### 2.2. DNA Manipulation

Primers used in the present study were listed in Table 2.

To construct pFA6a-TAP-cgLEU2, a DNA fragment amplified by PCR using primers #3964 and #3965 with pUC18-cgLEU2 (the National Bio-Resource Project, Shizuoka, Japan) was digested with *Asc*I and inserted between *Asc*I and *Sca*I in pFA6A-TAP-His3MX.

To construct pFA6a-GFP (S65T)-cgLEU2, a DNA fragment amplified by PCR using primers #3510 and #3971 with pFA6a-TAP-cgLEU2 was digested with *Xho*I and *Asc*I and ligated with a DNA fragment encoding GFP (S65T), which was amplified by PCR using primers #3212 and #3213 with pFA6a-GFP (S65T)-His3MX [5] and then digested with *Xho*I and *Asc*I.

To construct pFA6a-Venus-cgLEU2, a DNA fragment encoding Venus, which was amplified by PCR using primers #3212 and #3213 with pBS7 [10], was digested with *Xho*I and *Asc*I and inserted between *Xho*I and *Asc*I in pFA6a-GFP (S65T)-cgLEU2. The pBS7 was a gift from Eric Muller (Addgene plasmids #83774; http://nt2/addgene:83774; RRID:Addgene_83774, 21 December 2019).

To construct pFA6a-YFP-cgLEU2, a DNA fragment encoding YFP, which was amplified by PCR using primers #3212 and #3213 with pDH5 [10], was digested with *Xho*I and *Asc*I and inserted between *Xho*I and *Asc*I in pFA6a-GFP (S65T)-cgLEU2. The pDH5 was a gift from Eric Muller (Addgene plasmids #83775; http://nt2/addgene:83775; RRID:Addgene_83775, 21 December 2019).

To construct pFA6a-mCherry-cgLEU2, a DNA fragment encoding mCherry, which was amplified by PCR using primers #3671 and #3672 with pmCherry-C1 (Takara Syuzo, Shiga, Japan), was digested with *Xho*I and *Asc*I and inserted between *Xho*I and *Asc*I in pFA6a-GFP (S65T)-cgLEU2.

To construct pFA6a-13×Myc-cgLEU2, a DNA fragment encoding 13×Myc, which was amplified by PCR using primers #3478 and #3970 with pFA6a-13×Myc-kanMX [5], was digested with *Xho*I and *Asc*I and inserted between *Xho*I and *Asc*I in pFA6a-GFP (S65T)-cgLEU2.

To construct pFA6a-6×HA-cgLEU2, a DNA fragment encoding 6×HA, which was amplified by PCR using primers #3681 and #36 with pHyg-AID*-6×HA [11], was digested with *Xho*I and *Asc*I and inserted between *Xho*I and *Asc*I in pFA6a-GFP(S65T)-cgLEU2. The pHyg-AID*-6×HA was a gift from Helle Ulrich (Addgene kit #1000000120).

To construct pFA6a-ECFP-cgLEU2, a DNA fragment encoding ECFP, which was amplified by PCR using primers #4049 and #4050 with genomic DNA prepared from the BY25730 yeast strain (the National Bio-Resource Project, Japan), was digested with *Xho*I and *Asc*I and inserted between *Xho*I and *Asc*I in pFA6a-GFP(S65T)-cgLEU2.

DNA for plasmid construction was generated by PCR using Q5 DNA polymerase (New England Biolabs, Ipswich, MA, USA) according to manufacturer instructions. The *Escherichia coli* strain DH5α and standard media and methods were used for plasmid manipulation [12]. Genomic DNA was isolated as described previously by Hoffman and Winston [13]. PCR was performed to select clones with the correct gene replacements using Quick TAQ HS DyeMix (Toyobo, Osaka, Japan).

**Table 2 microorganisms-09-02505-t002:** Primers used in the present study.

Primer Name	Sequence (5′–3′)
#31	GTGTTTCTAGTGCTCCAATCAAAG
#36	CCGAATTCGTTGTTTATGTTCGGATGTGAT
#1206	CTATTATTGATGCTTTGAAGACCTCCAG
#3212	AACTCGAGGCGGCCGCTAACAGTAAAGGAGAAGAACTTTTC
#3213	TTGCATGCATATTACCCTGTTATCCCTAGC
#3478	AACTCGAGGCGGCCGCCGGGTTAATTAAGG
#3510	TTGCGGCCGCTCTGCAGAAGAAAATCTCATCCTCCGG
#3533	GAATTCGAGCTCGTTTAAAC
#3671	AACTCGAG GCGGCCGCTAACGTGAGCAAGGGCGAGGAGGA
#3672	AAGGCGCGCCCTACTTGTACAGCTCGTCCATGC
#3681	CTCAAAATGTCTCTCGAGGCGGCCGCCATCTTTTACCCATACGAT
#3964	GGGGGCGCGCCGTCTTTTTCAGTAATTTTCTTCCT
#3965	ACTGCATGCACATCTGCATCTTTAGAACCATCATA
#3970	CCCGGCGCGCCCTAGTGATTGATTAATTTTTGTTC
#3971	CCCCTCGAGTTTCCATCTTCTCTTCCATGGATTAA
#4049	GCTTAATTAAGCTCGAGATGGTGAGCAAGGGCGAG
#4050	TTGGCGCGCCATTACTTGTACAGCTCGTCCAT

Primer #31 was originally synthesized as a mutation primer to direct A1990G and G1991C (ATG = +1) mutations but can also be used to amplify *SWI5* [14] and correctly screen clones that have switched the C-terminal TAP tag to different epitopes.

### 2.3. Synchronous Cultivation from Metaphase via the GALp-CDC20 Method

Cells grown at 25 °C in SRaffGal medium to an OD_600_ of ~0.5 were pelleted, washed three times with H_2_O, and then incubated in SRaff medium to repress *CDC20* expression. When more than 90% of the cells showed a dumbbell-like morphology, galactose was added to the medium to activate *CDC20* expression and release the cells from metaphase arrest. Samples were collected at various time points for performing fluorescence microscopy.

### 2.4. Immunoblotting

Yeast cell lysates were prepared as described previously [15], electrophoresed, and immunoblotted with either anti-HA (12CA5, Roche, Basel, Switzerland), anti-Myc (9E10, Roche), or anti-TAP (Invitrogen, Walthan, MA, USA) as the primary antibody, and either anti-mouse (BioRad, Hercules, CA, USA) or anti-rabbit (Amersham Life Science, Amersham, UK) as the secondary antibody. A Lumi VisionPro HSII image analyzer (Aishin Seiki, Aichi, Japan) was used to visualize the immunoreactive bands.

### 2.5. Fluorescence Microscopy

Yeast cells were grown in SD or SRaffGal medium at 25 °C, and aliquots were collected for fluorescence microscopy. Microscopic analysis was performed using an Olympus IX83 inverted microscope. The following filters were used: GFP-3035-D-OFF (Semrock, West Henrietta, NY, USA), YFP-2427-B-OFF (Semrock), U-FCFP (Olympus, Tokyo, Japan), and U-FMCHE (Olympus; for mCherry).

## 3. Results and Discussion

We constructed a new set of plasmids that can provide modules for the replacement of the TAP tag in commercially available TAP strains with different tags. Modules for homologous recombination can be isolated by the digestion of their respective plasmids with the restriction enzymes *Sal*I and *Sac*I. All of these modules were based on MX6 and are illustrated in Figure 1a.

Furthermore, we made several modifications to the plasmids developed by Sung et al. [7]. First, as a 5′ recombination site, we extended the 26-nucleotide sequence used by Sung et al. [7] to 53 nucleotides using the DNA sequence from the *Sal*I to *Xho*I sites in pFA6a-TAP-His3MX (Figure 1b), and this change is expected to increase the efficiency of the targeted recombination. Notably, similar to the system developed by Sung et al. [7], a ~230 bp sequence of the *Ashbya gossypii TEF* terminator was used as a 3′ recombination site. Second, we used *cgLEU2* as a transformation marker instead of *KIURA3*. Therefore, we replaced *KIUARA3* and the *TEF* promoter with *cgLEU2* and its own promoter. This alteration would be useful in experiments where *URA3*-based plasmids are used. Finally, to inhibit recombination at the *ADH1* terminator located immediately downstream of the epitope-coding sequence, the sequence was replaced with the *Candida glabrata* CAGLoHo3773g terminator.

We first tested the conversion of the TAP tag in TK-691 with 13×Myc or 6×HA. The pFA6a-13×Myc-cgLEU2 and pFA6a-6×HA-cgLEU2 were digested with *Sal*I and *Sac*I, and the reaction mixtures were directly used to transform TK-691 cells. From 100 ng of each of the digested plasmids, 44 (from pFA6a-13×Myc-cgLEU2) or 74 (from pFA6a-6×HA-cgLEU2) transformants showed the Leu^+^ phenotype. To select clones with the correct gene replacements, we selected 10 clones from each transformant and screened them with the Leu^+^ His^−^ phenotype (Table 3). In each case, nine clones showed the Leu^+^ His^−^ phenotype, whereas one clone displayed the Leu^+^ His^+^ phenotype. These clones were further tested for correct gene replacement by PCR with one primer that matched with *SWI5* (#31) and others with the R1 core sequence (#3533) (Figure 1c). All cells with the Leu^+^ His^−^ phenotype produced DNA fragments with sizes ranging between 3 and 4 kbp (expected size: 3508 bp for Swi5-13×Myc; 3175 bp for Sw5-6×HA). These results indicate the successful switch of the TAP tag to the 13×Myc and 6×HA epitopes. We named the strains TK-1923 (*SWI5-13×Myc::cgLEU2*) and TK-1924 (*SWI5-6×HA::cgLEU2*).

The TK-1923 and TK-1924 strains were grown to the logarithmic phase at 30 °C, and cell lysates were prepared for immunoblotting. We detected Swi5-13×Myc and Swi5-6×HA in the lysates from the TK-1923 and TK-1924 strains, respectively, whereas Swi5-TAP was not detected (Figure 2). These results demonstrate that Swi5-TAP was switched to Swi5-13×Myc and Swi5-6×HA, and that 13×Myc and 6×HA were functional in the cells.

Next, we tested the conversion of the TAP tag in *GAL1p-CDC20 SWI5-TAP* cells (TK-1508) with a GFP (S65T) tag. The strain already has the MX6-based *GAL1p: kanMX* module to replace the authentic *CDC20* promoter with the galactose-inducible *GAL1* promoter. Since *CDC20* is essential for a metaphase-to-anaphase transition, growth depends on galactose. We used *GAL1p-CDC20* cells because synchronous release from metaphase arrest can be easily performed in these cells, thereby allowing easy tracking of changes in Swi5 localization during the cell cycle. The pFA6a-GFP (S65T)-cgLEU2 was digested with *Sal*I and *Sac*I, and ~100 ng of the reaction mixture was used to directly transform the TK-1508 cells. Transformants (23 colonies) that showed the Leu^+^ phenotype were further tested for the His^−^ phenotype. The ten transformants selected all showed the Leu^+^ His^−^ phenotype (Table 3). These clones were further tested for correct gene replacement by PCR, as described above. All clones that showed the Leu^+^ His^−^ phenotype produced DNA fragments with a size of 3–4 kbp (expected size: 3694 bp). These results indicate that the gene replacements were accurate. We named this strain TK-1527 (*GAL1p-CDC20 SWI5-GFP::cgLEU2*).

We followed the localization and degradation of Swi5-GFP (S65T) using fluorescence microscopy (Figure 3a). Swi5 changes its localization during the cell cycle; it is expressed in the M phase and retained in the cytoplasm. Contrastingly, it is translocated to the nucleus upon mitotic exit [16], followed by degradation via the SCF^Cdc4^ ubiquitin ligase [17]. We used the *GALp-CDC20* method for metaphase arrest [15]. We cultivated TK-1527 cells grown in SRaffGal medium to an OD_600_ of ~0.5, washed the cells three times with H_2_O, and cultured them again in SRaff medium to repress the expression of *CDC20* and arrest the cultured cells in their metaphase. Since *CDC20* is essential for the metaphase-to-anaphase transition, galactose removal results in cell cycle arrest in the metaphase. Subsequently, we released the cells from metaphase arrest by adding galactose to the medium and further culturing them for 30 min. As expected, fluorescence microscopy identified a GFP signal in the cytoplasm at t = 0 min, indicating the cytoplasmic localization of Swi5-GFP (Figure 3a). Strong signals of Swi5-GFP being in the nucleus were observed at 10 and 15 min timepoints afterTK-1527 cells were released from metaphase arrest (Figure 3a). At 20 min, most cells had Swi5-GFP in their nuclei, albeit with reduced levels. Most of the Swi5-GFP signals were invisible after 30 min (Figure 3a).

These results are in agreement with the localization of Swi5 during the cell cycle, as observed by indirect immunofluorescence [17], indicating that the GFP introduced into Swi5 is functional.

Finally, we converted Swi5-TAP to Swi5-Venus or Swi5-YFP, and Pma1-TAP to Pma1-mCherry or Pma1-CFP. The pFA6a-Venus-cgLEU2, pFA6a-YFP-cgLEU2, pFA6a-mCherry-cgLEU2, and pFA6a-ECFP-cgLEU2 were digested with *Sal*I and *Sac*I and transformed with TK-691 or TK1568. The number of transformants and the number of cells with Leu^+^ His^-^ or Leu^+^ His^+^ phenotypes are shown in Table 3. PCR analysis confirmed that the TAP tag efficiently switched to the Venus, YFP, mCherry, or ECFP epitope. These strains were named TK-1925 (*SWI5-Venus::cgLEU2*), TK-1926 (*SWI5-YFP::cgLEU2*), TK-1614 (*PMA1-mCherry::cgLEU2*), and TK-1927 (*PMA1-CFP::cgLEU2*), respectively. Fluorescence microscopy also confirmed this finding. The expression of Swi5-Venus and Swi5-YFP in the nuclei was evident in early G1 cells, whereas the fluorescence signals of Swi5-Venus and Swi5-YFP were not observed in small-budded cells (Figure 3b) which are indicative of an early S phase. Similarly, the localization of Pma1-mCherry and Pma1-CFP was examined using fluorescence microscopy. Pma1 is the major plasma membrane H^+^-ATPase [18] and is more abundant in ageing mother cells than in the daughter cells [19]. Our results confirmed the asymmetric distribution of Pma1 (Figure 3c): Pma1 levels in the plasma membranes were higher in the mother cells than in the daughter cells. These results indicate that the Venus, YFP, mCherry, and CFP epitopes are functional.

## 4. Conclusions

We herein constructed a set of plasmids that can be used to switch the MX6 module-based C-terminal TAP tags to other tags that include 13×Myc, 6×HA, GFP, Venus, YFP, mCherry, and CFP. The most notable feature of this method is its simplicity. Modules for epitope switching can be obtained by digestion of the plasmids with *Sal*I and *Sac*I. Reaction mixtures can be used to directly transform cells without purification. There is also no need to amplify modules using PCR. Another feature is the efficiency and accuracy of the devised method. Our results indicate that the DNA-sequence identities of 53 and ~230 nucleotides are sufficient for efficient and accurate homologous recombination. More than 90% of the transformants that exhibited the Leu^+^ His^−^ phenotype accomplished the correct epitope switch. Extension of the 26-nucleotide sequence used by Sung et al. [7] to 53 nucleotides increased the efficiency of the homologous recombination. In addition, we expanded the epitope switch system developed by Sung et al. [7]. Their system uses *KIURA3* as the transformation marker. Therefore, their system cannot be used in experiments that require plasmids with the *URA3* marker. Our plasmids use *cgLEU2* as a transformation marker; therefore, they can be used in experiments that use *URA3* as the marker. Furthermore, similar to the system developed by Sung et al. [7], we found that our system could be used to switch epitopes that already have additional MX6 modules. We showed an epitope switch in *GAL1p-CDC20::kanMX* cells. Taken together, our findings show that our plasmids could serve as useful tools for understanding the protein functions in yeasts.

## Figures and Tables

**Figure 1 microorganisms-09-02505-f001:**
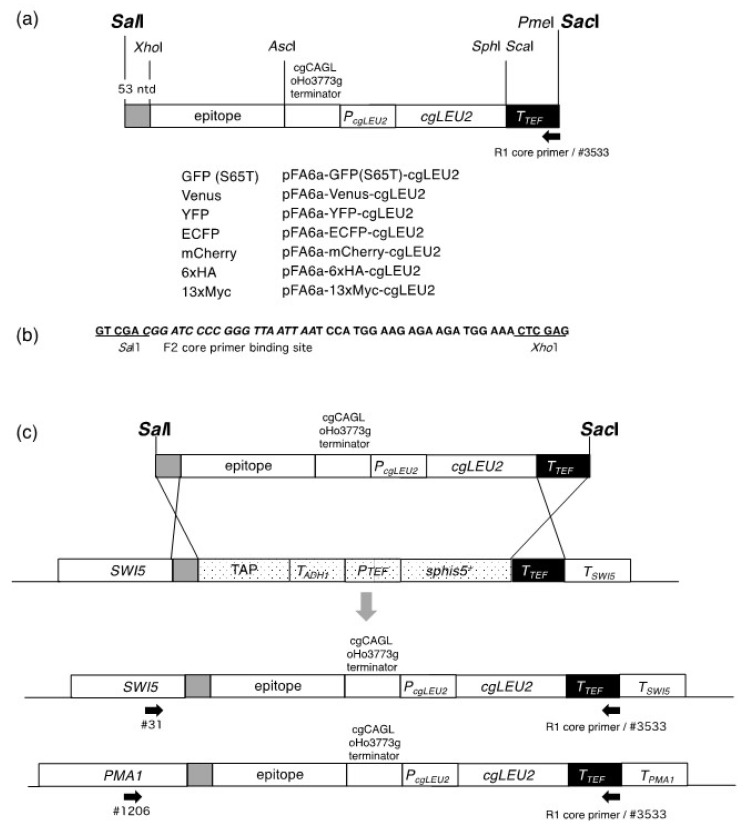
Modules for a C-terminal epitope switch and a schematic representation of the C-terminal TAP tag switch to various epitopes. (**a**) Modules for a C-terminal epitope switch. The modules for transformation can be excised by digestion with *Sal*I and *Sac*I. A homologous recombination site at the 5′ end is provided by the 53-nucleotide sequence from the *Sal*I to the *Xho*I sites that is common to the pFA6a series of plasmids (grey box). A 3′ homologous recombination site is provided by the *Ashbya gossypii TEF* terminator sequence (black box). The positions of the DNA sequences coding for epitopes, the *Candida glabrata* CAGLoHo3773g terminator, and the *cgLEU2* promoter and coding sequences are also shown. The arrow indicates the binding sites of the R1 core primer (GAATTCGAGCTCGTTTAAAC). The sequence of #3533 is identical to that of the R1 core primer. (**b**) Sequence comprising 53 nucleotides for the 5′ homologous recombination. (**c**) Schematic representation of the switch of the C-terminal TAP tag to various epitopes at the *SWI5* or *PMA1* locus. *Sal*I and *Sac*I were used to isolate modules for transformation and recombination. The positions of primers #31, #1206, and #3533 are shown.

**Figure 2 microorganisms-09-02505-f002:**
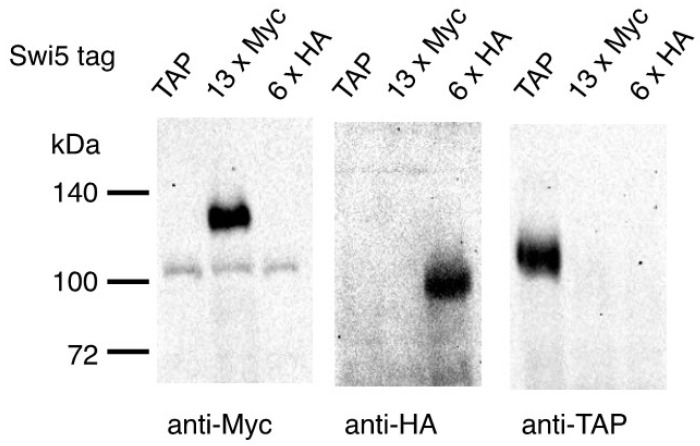
Expression of Swi5-13×Myc and Swi5-6×HA. Extracts from TK-691 (lanes indicated as TAP), TK-1923 (lanes indicated as 13×Myc), and TK-1924 cells (lanes indicated as 6×HA) were blotted with antibodies against TAP, Myc, and HA.

**Figure 3 microorganisms-09-02505-f003:**
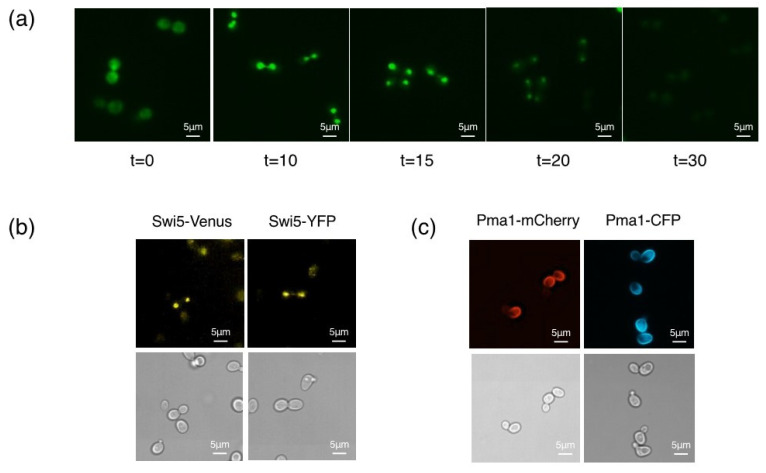
Fluorescence microscopic analysis of the expression of Swi5-GFP, Swi5-Venus, Swi5-YFP, Pma1-mCherry, and Pma1-CFP. (**a**) The *GAL1p-CDC20::kanMX SWI5-GFP::cgLEU2* (TK-1527) cells were arrested during the metaphase using the *GAL1p-CDC20* method, and the expression and localization of Swi5-GFP were determined at various intervals following their release from cell-cycle arrest using fluorescence microscopy. (**b**) The *SWI5-Venus::cgLEU2* (TK-1925) cells and *SWI5-YFP::cgLEU2* (TK-1926) cells were grown at 25 °C to the logarithmic phase, aliquots were collected, and the expression and localization of Swi5-Venus and Swi5-YFP were examined. (**c**) The *PMA1-mCherry::cgLEU2* (TK-1614) and *PMA1-CFP::cgLEU2* (TK-1927) cells were grown at 25 °C to the logarithmic phase, aliquots were collected, and the expression and localization of Pma1-mCherry and Pma1-CFP were examined.

**Table 1 microorganisms-09-02505-t001:** *Saccharomyces cerevisiae* strains used in the present study.

Strain Name	Genotype	Source
BY4741	*Mata ura3Δ* *0 leu2Δ* *0 his3Δ* *1 met15Δ* *0*	Euroscarf
TK-691	*Mata ura3Δ0 leu2Δ0 his3Δ1 met15Δ0 SWI5-TAP::His3MX*	Horizon
TK-1508	*Mata ura3Δ0 leu2Δ0 his3Δ1 met15Δ0 SWI5-TAP::His3MX GAL1p-CDC20::kanMX*	This study
TK-1527	*Mata ura3Δ0 leu2Δ0 his3Δ1 met15Δ0 SWI5-GFP::cgLEU2 GAL1p-CDC20::kanMX*	This study
TK-1568	*Mata ura3Δ0 leu2Δ0 his3Δ1 met15Δ0 PMA1-TAP::His3MX*	Horizon
TK-1614	*Mata ura3Δ0 leu2Δ0 his3Δ1 met15Δ0 PMA1-mCherry::cgLEU2*	This study
TK-1923	*Mata ura3Δ0 leu2Δ0 his3Δ1 met15Δ0 SWI5-13*×*Myc::cgLEU2*	This study
TK-1924	*Mata ura3Δ0 leu2Δ0 his3Δ1 met15Δ0 SWI5-6*×*HA::cgLEU2*	This study
TK-1925	*Mata ura3Δ0 leu2Δ0 his3Δ1 met15Δ0 SWI5-Venus::cgLEU2*	This study
TK-1926	*Mata ura3Δ0 leu2Δ0 his3Δ1 met15Δ0 SWI5-YFP::cgLEU2*	This study
TK-1927	*Mata ura3Δ0 leu2Δ0 his3Δ1 met15Δ0 PMA1-CFP::cgLEU2*	This study
BY25730	*Mata ura3-1 leu2-3,112 his3-11,15 trp1-1 ade2-1 SHS1-CFP::URA3*	NBRP

**Table 3 microorganisms-09-02505-t003:** Number of transformants with correct gene replacements.

		Numbers of Transformants				
Starting Strains	Epitope Switched	CFU/100 ng ^1^	Screened	Leu^+^ His^−^	Leu^+^ His^+^	Genomic PCR Passed
*SWI5-TAP::His3MX*	13×Myc::cgLEU2	44	10	9	1	9
	6×HA::cgLEU2	74	10	9	1	9
	GFP::cgLEU2	23	10	10	0	10
	Venus::cgLEU2	52	10	10	0	10
	YFP::cgLEU2	50	10	10	0	10
*PMA1-TAP::His3MX*	mCherry::cgLEU2	130	10	10	0	10
	CFP::cgLEU2	112	10	10	0	10

^1^ CFU/100 ng: colony forming units per 100 ng of plasmids digested with *Sal*I and *Sac*I.

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
