# Peer review of "A Set of Plasmid-Based Modules for Easy Switching of C-Terminal Epitope Tags in Saccharomyces cerevisiae"

_microorganisms, 2021, doi:10.3390/microorganisms9122505_

Round 1

Reviewer 1 Report

Saccharomyces cerevisiae strains, whose open reading frames are tagged with different epitopes, are widely used at present and many of them are commercially available. The manuscript ‘microorganisms-1465655’ describes a set of plasmids bearing restriction enzyme excisable modules designed for replacement of the TAP tag in such strains for another tag. This set includes 13xMyc, 6xHA, GFP, Venus, YFP, mCherry, and CFP tags. Thus, this set of constructs seems to be very useful and might be  implemented by many researchers if these plasmids would be freely available. 

  1. Suggestion for the title: A set of plasmid-based modules for easy switching C-terminal epitope tags in Saccharomyces cerevisiae
  2. A similar approach has been used previously (PMID: 31285266, DOI: 10.1242/bio.044529). The authors are unlikely to know about this, since this has not been mentioned in the abstract of that paper and was only briefly described in the Materials and Methods. However, it would be reasonable for the authors to reference this paper.
  3. Lines 42-44: Usage of the word 'conventional' is not clear. Please, use another expression.
  4. line 43: It seems that the term ‘homology’ should be used instead of ‘homologous’
  5. In many cases 'Sung et al.' is not followed by a corresponding reference.
  6. lines 52-53: This sentence sounds imperfect. Possibly it could be modified into ‘In the present study, we constructed a set of plasmids bearing modules for epitope switching, which can be generated by restriction digestion’.
  7. line 108. The subheading is not sufficiently informative.

  1. Figure 1. The F2 primer is mentioned only in this figure. Its purpose is not clear. This should either be removed or explained in more detail.

  1. Lines 160, 162: Designations SD-His and SD-Leu are misleading. It is unclear whether there is minus or hyphen. The corresponding data can be explained as His+ and Leu+ phenotypes.
  2. Figure 2 hampers rather than helps understanding of what is described in the text. Indeed, it mentions 10 transformants in the text with reference to their analysis in Figure 2. The spots and gel lanes are also numbered from 1 to 10 in the figure. However, it becomes clear that numbers in the figure does not correspond to numbers of the transformants only after careful reading of the figure legend. Actually, this figure appears not to be required for the article. It should either be removed or presented in a more appropriate manner (possibly as a supplement).
  3. line 164 and the same cases through the text: Term 'genomic PCR' sounds strange. It can be replaced for just 'PCR' without loss/change of meaning.
  4. Sentence starting at line 189 is not required.

13 Figure 3: Numbering of lanes is not convenient for the reader. Possibly, the lanes could be marked by the Swi5 tag: 13xMic, 6xHA, and TAP. The meaning of 'alpha' is unclear; the word 'anti' in this case could better indicate that these are antibodies against corresponding epitopes. The antibody codes are not required in the legend since they are presented in the Materials and Methods.

  1. Paragraph starting at line 198: The aim of the use GAL1p-CDC20 strain has not been explained. The experiment, which requires the use of such mutant, is described in the following paragraph. However, what this experiment provides to the general idea of the manuscript is not clear.

  1. The sentences at lines 241-243 'Genomic PCR results of cells with the Leu+ His- phenotype are shown in Figure 2b. These results indicated that the …' could be combined in a phrase like this: 'PCR analysis confirmed that the TAP tag efficiently switched to the Venus, YFP, mCherry, or CFP epitope (Figure 2B).' The reference to a figure is not necessary, since such analysis is a quite routine procedure and Figure 2 can be removed completely (see also the comment on figure 2 itself).
  2. line 258: Term 'outstanding' sounds overly optimistic in this case. Possibly, something like 'notable' would be better.
  3. A phrase about plasmid availability would encourage more researchers to request and use the tools created in this paper.

Reviewer 2 Report

The manuscript deals with the fairly simple construction of plasmid containing epitopes that could be used for tagging genes in lab strains of S. cerevisiae. In general the manuscript is concise and not badly written but there are some issues that need to be addressed before it could be accepted for publication.

-the M&M are sometimes lacking, in some cases I assume E coli was used for plasmid propagation thus could you mention it. In addition, simply just mention what Taq polymerase/reaction kit was used as it was important for plasmid construction as well as confirmation of transformants

-the photos are sometimes not clear and are all missing a bar to indicate the resolution. Especially Fig 4a where it is not clear that the protein migrated to the nucleus. The signals just seems less intense.

-are these plasmids deposited somewhere like addgene or something similar?

-Of concern is the primer sequences. I had a look at the first two in Table 2 #31 and #36 to see if they aligned with the sequence for the gene obtained from SGD. #31 has got two nucleotides that are not the same but basically half of #36 does not align with the sequence from SGD. For a manuscript like this which is largely about plasmid/strain construction it is important that the primer sequences should be correct as people might really like to replicate it and thus if the first two are already not correct, it cast doubt on all the others. Please clarify how the primers would have annealed as #36 would definitely not

For SWI5

GTGTTTCTAGTAGTCCAATCAAAG

GTGTTTCTAGTGCTCCAATCAAAG

For PMA1

CCGAATTCGTTGTTTATGTTCGGATGTGAT

GTGAAAGAATTGTTTGTGTTAAGGGTGCTC

Reviewer 3 Report

Nice manuscript showing that these plasmids can serve as a tool for understanding protein functions in yeasts. I recommend this manuscript for publication in Microorganisms.

Author Response

Thank you very much for highly evaluating our manuscript, and recommending it for publication in Microorganisms.